# Privacy Engineering for Domestic IoT: Enabling Due Diligence

**DOI:** 10.3390/s19204380

**Published:** 2019-10-10

**Authors:** Tom Lodge, Andy Crabtree

**Affiliations:** School of Computer Science, University of Nottingham, Nottingham NG7 2RD, UK

**Keywords:** general data protection regulation (GDPR), data protection by design and default (DPbD), data protection impact assessment (DPIA), due diligence, privacy engineering, internet of things (IoT), databox, integrated development environment (IDE)

## Abstract

The EU’s General Data Protection Regulation (GDPR) has recently come into effect and insofar as Internet of Things (IoT) applications touch EU citizens or their data, developers are obliged to exercise due diligence and ensure they undertake Data Protection by Design and Default (DPbD). GDPR mandates the use of Data Protection Impact Assessments (DPIAs) as a key heuristic enabling DPbD. However, research has shown that developers generally lack the competence needed to deal effectively with legal aspects of privacy management and that the difficulties of complying with regulation are likely to grow considerably. Privacy engineering seeks to shift the focus from interpreting texts and guidelines or consulting legal experts to embedding data protection *within* the development process itself. There are, however, few examples in practice. We present a privacy-oriented, flow-based integrated development environment (IDE) for building domestic IoT applications. The IDE enables due diligence in (a) helping developers reason about personal data during the actual in vivo construction of IoT applications; (b) advising developers as to whether or not the design choices they are making occasion the need for a DPIA; and (c) attaching and making available to others (including data processors, data controllers, data protection officers, users and supervisory authorities) specific privacy-related information that has arisen during an application’s development.

## 1. Introduction

The European Union’s General Data Protection Regulation (GDPR) [1] has recently come into effect to strengthen the protections provided to ‘data subjects’ or citizens (end users in more technical terms) in the face of “rapid technological developments”. It is targeted at any system physical or digital that ‘processes’ (collects, structures, stores, adapts, distributes, destroys, etc.) personal data and is global in reach insofar as the processing touches EU citizens or their data no matter where it takes place. GDPR defines a set of actors to which compliance applies, including ‘data controllers’ (parties who commission and determine the purposes of personal data processing) and ‘data processors’ (parties who process data on the controller’s behalf, including application developers that actually enable data processing). Core to compliance is a set of Data Protection by Design and Default (DPbD) principles such as data minimization and informed consent that are aimed at instilling good practice and reducing risks to the data subject’s rights and freedoms.

Whether application developers like it or not, they are now obliged by regulation to take DPbD into account and exercise due diligence unless they wish to be implicated in legal action that may result in fines of up 10 million euros or 2% of worldwide annual turnover, whichever is greater and which may be doubled if non-compliance is not rectified on the order of an EU member state. Application developers can no longer treat data protection as someone else’s problem, a disconnected process that sits solely within the province of legal specialists. Rather, they will either be directly accountable as data controllers or will be indirectly implicated by a requirement to ensure that data controllers are able to fulfil their obligations in their capacity as data processors or actors who enable the data processing operations.

A key heuristic mandated by GDPR to enable due diligence is the Data Protection Impact Assessment (DPIA). DPIAs are a formal assessment of the privacy risks posed by a system; they are mandatory in cases where processing is likely to be ‘high risk’ and are otherwise recommended wherever personal data is processed. Both the assessment of whether a DPIA is necessary and the act of conducting a DPIA will draw heavily upon the details of implementation, even if developers are not directly accountable for performing a DPIA. The EU’s A29WP or Article 29 Data Protection Working Party (now the European Data Protection Board) is clear on this: “If the processing is wholly or partly performed by a data processor, the processor should assist the controller in carrying out the DPIA and provide any necessary information” [2]. Compliance with privacy regulation has therefore become a mundane but necessary feature of application development.

As Roeser [3] notes: “Engineers can influence the possible risks and benefits more directly than anybody else.” However, despite the promise, research has shown that developers of all shades and hues generally lack the competence to deal effectively with privacy management [4,5,6,7] and that the difficulties of complying with regulation are likely to grow considerably [8]. Traditionally, developers are faced with a couple of options. Firstly, they may hand off due diligence to experts. This presupposes that (a) developers have access to such experts, which is unlikely in the case of a lone developer or small business, and (b) the privacy expert has the technical competence needed to understand the impact of development decisions upon privacy. Of course, guidelines exist, yet these tend to be oriented towards a legal rather than technical domain. When they are aimed at the developer, in the form of privacy patterns [9], checklists [10] and frameworks [11,12,13], they remain disconnected from the tools and environments developers use to build their applications. This can mean that they are employed in the later stages of the development workflow, rather than early on when changes to an application to improve upon privacy are most easily and effectively made. Moreover, developers are faced with a challenge of relating guidance to concrete technical considerations, i.e., whether a particular design choice or implementation detail appropriately reflects given advice. 

By way of contrast, the discipline of ‘privacy engineering’ [14] has recently emerged and aims to embed privacy practice more deeply in the actual development process. Privacy engineering practices are derived from privacy manifestoes such as privacy by design [15] and regulation (such as GDPR) and may include data flow modelling, risk assessment and privacy impact assessments. Good privacy engineering aims to embed privacy practice seamlessly into a developer’s workflow, to allow it to be more easily assessed in relation to the specific artefacts being designed and implemented. By making privacy tools a core part of the development process, rather than something that sits alongside it, developers can be sensitized to the use of personal data (i.e., what it is and how it is being exploited) during the actual in vivo process of building applications. This, in turn, may encourage developers to reason about the implications of their applications and explore alternatives that minimize the risks of processing personal data. More concretely, by building notions of privacy into a development environment, the environment itself may provide guidance that relates specifically to the artefacts being built and can even feed into formal processes required by GDPR, such as the DPIA, to enable due diligence.

This paper contributes to current efforts to engineer privacy through the design of an integrated development environment (IDE) enabling due diligence in the construction of domestic IoT applications. The IDE enables due diligence in (a) helping developers reason about personal data during the in vivo construction of IoT applications; (b) advising developers as to whether or not the design choices they are making occasion the need for a DPIA; and (c) attaching and making available to others (including data processors, data controllers, data protection officers, users and supervisory authorities) specific privacy-related information about an application. Our environment focuses on the development of domestic IoT applications as these present a relevant and interesting privacy challenge, operating in traditionally private settings and commonly processing data that is personal in nature (though not, as we shall see, always immediately and obviously so).

## 2. Background

In this section, we begin with an overview of the various legal frameworks and regulatory guidelines that impact application development that involves the processing personal data. Of particular relevance to our work are (a) Data Protection by Design and Default (DPbD), a legally mandated successor to Privacy by Design that provides a set of best practice principles and emphasizes early engagement with data protection, and (b) Data Protection Impact Assessments (DPIAs), a primary instrument and heuristic for assessing privacy and demonstrating compliance with the principles of DbPD. We follow this with an overview of the support currently available to developers, covering frameworks and guidelines and work on codifying and translating legal concepts to technical domains. We conclude with a summary of various technical privacy tools and APIs available to developers.

### 2.1. Legal Frameworks

**Data Protection by Design and Default.** DPbD expects engagement in privacy-oriented reasoning early on in the design process, rather than as a late stage bolt-on for post hoc compliance. It has been embraced by Data Protection Authorities [16,17], is supported by the European Commission [18,19] and is legally required by GDPR [1]. DPbD promotes various data protection measures such as data minimisation, pseudonymisation and transparency as well as appropriate system security. Although the legal requirement for DPbD ought to provide sufficient incentive for developers to engage with its principles, there is no evidence that it has yet gained widespread, active adoption in the engineering process. This may in part be due to the fact that its principles are largely disconnected from the actual practice of systems engineering; translating between the sometimes vague terminology in regulation to concrete engineering outcomes is non-trivial. Take, for example, the requirement for data minimisation. In practice, this can include: minimising data collection, minimising disclosure, minimising linkability, minimising centralisation, minimising replication and minimising retention [20].

**Data Protection Impact Assessment.** The DPIA is the recommended instrument for demonstrating compliance with regulation. However, DPIAs are not mandatory in all cases. This means that the decision as to whether or not a DPIA is required can only be made with reference to the detail of the design and implementation of code. DPIAs are legally required when a type of processing is “likely to result in a high risk to the rights and freedoms of natural persons” [1] While there is no definitive template for a DPIA, GDPR states that there must be “a level of rigour in proportion to the privacy risk arising” (ibid.). This can be a nuanced distinction, especially when apps run on privacy enhancing technologies (PETs) designed to offer privacy protection. Even when personal data from IoT devices are processed, the requirement for a DPIA may be waived if the app presents a low privacy risk to a data subject. All of this and more draws into question how a developer is to assess the need for a DPIA? Any effort must draw heavily upon context and consider the particular environment an app will operate in and how it will process personal data.

### 2.2. Documentation

**Frameworks.** Several consortia promote the use of frameworks that adopt a common lexicon and standardised definitions for privacy practice. OASIS’s Privacy Management Reference Model [10], for example, is an 11,000-word document that presents steps to identify risks and ways to address them. The National Institute of Standards and Technology Privacy Risk Management Framework [12] is an ongoing effort at draft stage and aimed at organisations to help “better identify, assess, manage, and communicate privacy risks.” It takes a harm-based perspective on privacy, identifying a range of potential harms and developing a risk management approach (including a set of specific privacy preserving activities) to address them. There are also a variety of frameworks used to help developers capture the development requirements necessary to mitigate likely privacy threats for a specific use case. Most notable is the LINDDUN methodology [13], which instructs developers on what issues should be investigated, and where in a design those issues could emerge. The methodology requires developers to construct use cases as Data Flow Diagrams (DFD) from a set of predefined elements including (a) an external entity (an endpoint of the system), (b) a process (a computational unit), (c) a datastore (data at rest), (d) dataflow (data on the move), and (e) a trust boundary (the border between untrustworthy and trustworthy elements). The resulting use cases can then be expanded into a set of high-level privacy engineering requirements, which are used as a basis for selecting from a set of privacy solutions to mitigate perceived threats (i.e., design patterns such as unlinkability, unobservability). The creation of the DFD is key to the analysis. Naturally, an incorrect DFD will lead to incorrect results.

**Ideation Cards.** Luger et al. [21] consider an alternative approach towards enabling developers to respond to privacy risks. Their human-centered approach focuses on the early design stage of development, i.e., from the minute that “pen hits paper”. They use a card game as an approach to sensitise and educate designers to regulation and its requirements, where the cards foreground designers’ reasoning about four core GDPR tenets: informed consent, data protection by design, data breach notifications, and the right to be forgotten. The study reveals how “a lack of awareness of the breadth of regulation” limits the “desire or ability to incorporate such considerations at an early stage.” In exploring ideation cards with a range of development practitioners (designers, programmers, human-computer interaction developers), Luger et al. express real concern over the threshold of knowledge required to understand and respond to regulation when designing and building applications that exploit personal data.

### 2.3. Bridging between Tools and Design

We turn now to consider the technical tools that are currently available to developers and offer a bridge between regulation and development practice.

**Policies/Ontologies**. GDPR, as with most legal texts, cannot be codified and subsumed into a purely technical system. Policy and law frequently prescribe non-computational decisions that can only be affected by human judgment. In spite of the inherent difficulties of formalising regulation, work has been undertaken to create translations between the legal and technical domains. Typically, this has involved the codification of legal text, to create formalisms that can be directly utilised by systems and code. The objectives vary from seeking to enforce legally binding policies [22,23] to assessing a system’s compliance or comparing the degree and scope to which systems comply with regulation. Pandit et al. [24], for example, provide an ontology for formalising relationships between regulations; Fatema et al. [22] present a method that uses Natural Language Processing to automate the translation between data protection rules and policy rules; and Corrales et al. [25] elaborate an alternative approach that requires legal specialists to work more directly with code. The authors describe a system for building smart contracts that uses pseudo code (in the form of a set of IF THEN clauses) written by lawyers, which can be subsequently translated into a contract. However, the authors goal is relatively modest—to ‘nudge’ cloud providers to a greater level of compliance by aligning their Service Level Agreements with respect to data processing more closely with GDPR.

**Design Patterns.** Privacy design patterns are repeatable solutions to recurring problems within a given context [9]. Patterns may be chosen and applied to satisfy a set of privacy goals, e.g., anonymity, unlinkability, unobservability, pseudonymity, minimisation, etc. In most cases, systems will require that a variety of patterns are composed together using a pattern language. In the context of privacy by design, a few design patterns have been presented to date. Hafiz [26], for example, presents a set of privacy mitigations in response to known attacks and extends the work to present a pattern language to help developers choose patterns that are relevant to a particular domain (for which the domestic IoT is not one) [27]. Pearson [28,29] argue that we need to move beyond guides and checklists and towards “automated support for software developers in early phases of software development” given that “developers do not usually possess privacy expertise”. The authors utilise a decision support-based system where developers provide a set of requirements (in the form of a questionnaire) and these are then used to generate a candidate set of high-level privacy patterns (such as opt-in consent, or negotiation around privacy preferences) that must be implemented by the developer. Privacy patterns offer a useful reference for developers [30], though typically at a relatively late stage of design when a range of choices and decisions have been made, and they help document common practices and standardize terminology. Practically, however, the problem remains that developers must have the necessary competence to choose and implement patterns relevant to the artefacts that they are designing.

### 2.4. Privacy Tools

The discipline of privacy engineering aims to improve support for developers by embedding privacy support more deeply into everyday development tools. Various tools have emerged to help developers build applications and systems that are compliant with (some aspect of) GDPR. These tools concentrate on *technical compliance* with GDPR; for example, that their systems meet security standards, that they restrict the flow of personal data, or that they fulfil contractual obligations and meet auditing requirements.

**Automated Program Analysis.** Some progress has been made in considering how program analysis techniques might be used to help demonstrate privacy compliance. The approach has its roots in efforts to assess where applications contain security vulnerabilities (e.g., data leakage, permission misuse, malware). Taint analysis, for example, examines program execution by tracking local variables and heap allocations. Sensitive data are labelled or ‘tainted’ so they can be tracked as they flow through a system and arrive at a ‘sink’ (the point where data is exposed, e.g., network access). Tainted data may move through multiple paths before arriving at a sink, so multiple flows may need to be analysed. An important consideration with taint analysis is identification of the sources of data that are deemed to be sensitive. In the mobile domain, the authors of Taintdroid [31] taint phone sensor data (e.g., microphone, accelerometer, GPS, camera), contacts and text messages and device identifiers (IMEI, phone number, SIM card IDs). The authors of Flowdroid [32] extend taints to include data from user interface fields (e.g., user passwords). In IoT domains, SAINT [33] marks all sensor device states, device information, location data, user inputs and state variables as sensitive. To make analysis tractable, there is a trade-off is between ensuring that all potentially sensitive data are tainted whilst minimising false positives that arise from tainting non-sensitive data. Ferrara [34,35] directly applies taint analysis to demonstrate compliance with GDPR. The outputs from analysis take the form of *privacy leak reports*. However, further contextually dependent analysis is required to determine whether a ‘leak’ actually constitutes a violation of privacy. While program analysis techniques can be used to demonstrate that an application is free of certain security vulnerabilities and inform analysis of privacy vulnerabilities, higher-level analysis is needed to identify, for example, where data exposure is a deliberate and legitimate design decision or, alternatively, where data marked as ‘sensitive’ is or is not so within a specific context. The specific intentions of an application, the specific data being processed, the specific processing being undertaken and the context of disclosure must all play a part in the analysis of taints.

**APIs and Services.** We are also seeing the emergence of cloud-based data privacy services or Data Privacy as a Service (DPaaS) that sit between applications and data providers. The types of services on offer include disclosure notices, consent management, compliance software or solutions to more technically challenging privacy engineering tasks. Privatar, for example, provides Publisher [36], which offers policy-controlled data pipelines to de-identify data, and Lens [37], which utilises differential privacy to provide insights over sensitive data. However, this class of solution is not aimed at the developer per se, but instead seeks to enable businesses to *outsource* privacy management.

## 3. Designing for Due Diligence

Despite the laudable aims of privacy engineering, we remain unaware of any examples of privacy tools that are actually embedded within development environments and allow developers to reason about the risks to privacy created by the specific artefacts that they are building as they are building them. In this section, we consider how in vivo support for due diligence might be more strongly embedded within development environments. We start by drilling down into Data Protection Impact Assessment before relating guidance to features provided by our own development environment.

GDPR promotes the use of a DPIA for two purposes: first, as a tool for evidencing compliance, and, second, as a heuristic device enabling data protection by default. As A29WP’s guidance on DPIAs [2] puts it “in cases where it is not clear whether a DPIA is necessary”, one should be “carried out nonetheless” as resolving the matter is key to complying with data protection regulation. Moreover, A29WP takes the view that “carrying out a DPIA is a continual process, not a one-time exercise” that will “encourage the creation of solutions which promote compliance” (ibid.). However, from the developer’s perspective, the more pressing concern might be to avoid punitive fines and to thus be able to demonstrate to oneself the parties one works for, or to the authorities that reasonable steps have been taken to comply with the regulation and the developer has thus acted with due diligence. The first consideration for a developer, therefore, is to determine whether or not a DPIA is actually necessary.

Article 35(1) of the General Data Protection Regulation, states that a DPIA must be carried out: “Where a type of processing in particular using new technologies, and taking into account the nature, scope, context and purposes of the processing, is likely to result in a high risk to the rights and freedoms of natural persons …”

To complicate matters, specific guidance on whether or not a DPIA is necessary falls under the jurisdiction of each EU member state (e.g., the UK’s Information Commissioner’s Office or ICO). However, A29WP provides general clarification, defining nine “common criteria” and states that “processing meeting two criteria would require a DPIA to be carried out.” The criteria include:Evaluation and scoringAutomated decision making with legal or similar significant effectWhen the processing prevents data subjects from exercising a right or using a service or entering a contractSystematic monitoringSensitive data or data of a highly personal natureMatching or combining datasetsInnovative use or applying new technological or organisational solutionsData processed on a large scaleData concerning vulnerable subjects

The first two criteria are concerned with the methods used to process personal data: *evaluation* and scoring relates to the use of personal data for the creation of models that might then be used as a basis for profiling or predicting; automated decision-making is concerned with outcomes generated without any human intervention with the principal risk being discriminatory algorithms that create legal or similarly significant effects. The third criteria cover any cases of processing that can result in a denial of service, right or contract (e.g., insurance, government services, education, finance). Systematic monitoring is concerned with the observation, monitoring or control of data subjects, particularly when they are unaware and have not provided consent; the predominant scenario is monitoring in public spaces where it may be impossible for individuals to avoid processing; however, domestic settings are not exempt from consideration given that sensors may often gather data without the explicit awareness of householders. In home environments, it is tempting to simply suggest that all data that originate from IoT devices are sensitive or of a highly personal nature as per criterion number five, given that it relates to a private setting and may be attributable to a household or individual. This would, in our view, impede a deeper engagement with the design implications of working with specific types of personal data. The risks involved in *matching and combining datasets* emerge from the potential for secondary inferences to be made without the explicit consent of a user. The guidance is primarily concerned with cases where the combined data is sourced from independent data processing operations or controllers (rather than, for example, within a single app) and outside the remit of the original purpose of processing. With respect to criteria number 7, innovative use or new technological or organisational solutions, there is a view that “due to the growing importance of the IoT and the associated risks for privacy, a DPIA will be mandatory for most IoT devices” [18]. However, this is arguably of limited utility as the IoT will not remain innovative for long and is indeed already becoming commonplace. Furthermore, we imagine that any diligent developer will want to assess the privacy implications of any new solution they create if only to avoid penalty. Criteria number eight, data processed on a large scale, can be interpreted as numbers of users, volume of data, geographic reach and/or the duration and permanence of the processing activity. Criteria number nine, data concerning vulnerable subjects, refers to data processing that relies on a power imbalance between data subjects and controllers and includes, for example, children, those with disabilities, workers, etc. While there is clearly a good degree of interpretive flexibility in determining just what the common criteria require or amount (see the ICO, for example [10]), they nevertheless provide a framework around which we can consider embedding due diligence within the actual in vivo process of building IoT applications. Thus, in the following section, we consider how we can exploit knowledge about the way an app is constructed and the method and type of data being processed to offer points for reflection and advice enabling developers to exercise due diligence.

## 4. Embedding Support for Due Diligence in Design

As underscored by A29WP, the question of whether or not a DPIA is required lies at the heart of due diligence. It is a question that must be continuously asked throughout development and so our aims in seeking to embed due diligence into an IoT application development environment are:To enable reflection and in vivo reasoning during app construction concerning the impact of the developer’s design choices on privacy risks;To provide advice to developers that responds to data protection regulation and which is specific to the application being built. This advice will relate:To determining whether or not a DPIA is likely to be needed;To the details of design/implementation that most influence that decision.


Our starting point is to consider an overall paradigm that suits both the development of IoT applications and reasoning about privacy. GDPR is principally concerned with the flow of personal data and it is common, when assessing data privacy, to employ data flow diagrams, where data flows are represented as edges between nodes. Nodes may be locations (e.g., EU member states or third countries), systems, organisations or data subjects. This is where reasoning about data protection must take place. Edges between nodes represent data transfer. Data flow modelling can also form the basis of programming environments, in particular data flow programming, which emphasises the movement of data between nodes along a series of edges. It is particularly suited to designing applications and systems that are composed of a set of ordered operations. With data flow programming, nodes are black boxes of functionality and edges provide connections over which data is transferred. This paradigm can be particularly effective when applied to IoT (see node-RED, ioBroker, no-flow, total.js), where it can be natural to model systems and applications as sources of data, connected to processes en route to a final endpoint (i.e., actuation or external service).

We therefore use a data flow modelling paradigm as the basis of our development environment. We leverage it to simultaneously build application functionality and support reasoning about privacy. In our case, we distinguish between several top-level node types that are straightforward to translate between privacy and developer domains. The simplest view of an application is of *sources* of data, processors that act upon the data, and sinks or *outputs* that produce an event (which may or may not include external data transfer). We consider the reasoning attached to each:Sources of data. When working with data, the first job of the developer is to determine which data is personal, and which data is not. Perhaps as a testament to how difficult this can be, the UK ICO has published a 23-page document titled ‘What is Personal Data?’ [38]. By understanding the data, a developer is able to reason about the fifth DPIA criterion (sensitive data or data of a highly personal nature) when judging whether or not a DPIA is necessary.Processors. When operating upon personal data, processors may reidentify a data subject, generate new personal data (using, for example, data fusion, inference, profiling) or remove personal data (e.g., through anonymisation). To relate the structure of apps more closely to regulation, in addition to a standard processor (what we call a transformer), we add two special classes: a profiler and a privacy pattern. A profiler will infer new personal information (not already explicitly contained in its input data) about a data subject. A privacy pattern is one of a set of ‘off the shelf’ privacy preserving techniques (e.g., pseudo/anonymisation, metadata stripping, encryption) developers may make use of to help resolve privacy issues. By making these distinctions, we can help developers reason about the first, second and sixth DPIA criteria (evaluation and scoring, automated decision making, matching or combining datasets). Machine learning is commonly utilised to infer new personal data and our profile nodes capture cases where this occurs within an app. Note that our intention is not to surface issues relating to the quality of the algorithms or models employed by profilers (such as concerns around fairness/adverse consequences) as this sits beyond the scope of our work. Rather, it is to offer prompts for reasoning, given the emphasis GDPR regulation places upon the use of automated profiling (Recital 71, Article 22(1)).Outputs. These represent the points where something is done by an application; it may be that sensors are actuated, and data are visualised, stored, or transferred from the app. It is here that careful reasoning about how data are used must occur, and the remaining DPIA criteria must be considered.

Figure 1 provides a summary of the components of our development environment. Applications can be constructed by creating connections between combinations of nodes. The assumption is that each node will contain core logic to provide a common operation which can then be specialised through configuration, but we also assume that there is a ‘general’ processor to allow developers to process data through raw code. The advantage of providing pre-baked logic, aside from simplifying development, is that the IDE can automatically calculate how the characteristics of data change after each processing operation. This enables developers to reason about data’s characteristics as it flows into an output (the point at which risks are realised).

### 4.1. Tracking Personal Data

What, then, are the characteristics of the data that we wish to capture? Referring back to our aims, we want to understand if the data is personal and/or sensitive in nature. GDPR distinguishes between ‘personal’ and ‘non-personal’ (anonymous) data, with a further distinction around ‘special categories’ (Article 9, GDPR) to which additional protections must apply. How might this be operationalised? How can we use it to get to the point where developers are able to reason about whether (often seemingly benign) data has the potential to reveal personal and even sensitive insights, or how new transformations (such as fusion with other sources, profiling and inference) will affect data’s personal characteristics? To tackle this challenge, we consider a minimum set of personal data features that will enable this reasoning:***Data categorisation.*** Underlying the need for categorisation is the recognition that all personal data is not equal. Article 17(1) and Recital 46, GDPR, confirm that the “nature of the personal data” to be protected should be taken into account. Craddock et al. [39] note that categorising personal data is one way of contextualising it and understanding its nature. They thus note that categories can act as an anchor to which further information about processing can be attached, which can help with the creation of Data Protection Impact Assessments. From the perspective of a developer, categorisation can act as an aid to reasoning around the nature and severity of risk for particular design decisions.***Derivable characteristics.*** Various combinations of personal data, under certain conditions, will lead to the possibility of further personal inferences. To this end, we distinguish between primary and secondary sources of personal data. Secondary data, as with primary, will correspond to a category but will also be attached to a set of conditions that must be met before the category is realised. Typically, these conditions might include the existence of additional categories of personal data (for example, given various physical characteristics it may be possible to infer gender), but that might also might refer to other conditions such as data sampling rate (live energy data may make it possible to derive house occupancy patterns whereas daily readings will not). There is a distinction between the possibility and actuality of an inference being made. That is, there will be many cases where a set of conditions make inferences theoretically possible, even if they are not being made during an app’s processing of data.***Data accuracy.*** Relatedly, the accuracy of a particular source of personal data can have as significant a bearing upon potential privacy harms as the data themselves. There is a challenge in ensuring that applications do not lead to unfair, unrepresentative, and even dangerous outcomes as a result of inaccurate data. Getting developers to reason about the accuracy of the data that their applications process is therefore a part of the mix.

With regard to data categorisation, various categories have been proposed in the literature [40], by data protection authorities [1] and technical consortia [41]. There is considerable divergence across approaches and weaknesses in each [39]. For our own purposes, none are expressive enough to capture the types of IoT data that developers are working with, nor are they able to support the particular form of developer reasoning around personal data processing that we require to enable due diligence. In developing our own categorisation, we are not presenting our work fait accompli, but as a vehicle for investigating how we might contextualise personal data to improve support for developer reasoning about privacy risks.

With regard to derivable characteristics, we want to be able to calculate and represent the creation or loss of personal data at each processing point within an application. Figure 2 provides a summary of each transformation we deal with. The blue squares represent a data processor that performs some kind of transformation on personal data inputs (Px). The conditions block (v) shows, in the first case, that when personal data is captured at a low frequency, it is not possible for an inference (*Pc*) to be made. In the second case, with a higher sampling frequency, the inference is made possible.

### 4.2. Developing a Categorisation

We now present more detail on the schema we have developed to enable tracking and reasoning around the flow of personal data within our privacy-aware IoT applications. Our schema distinguishes between three top level classes of personal data (identifier, sensitive, personal). It provides a set of subcategories and a set of subtypes under each subcategory. As a start, inspired by GDPR, we specify six top-level personal data types (Table 1). In our schema (Table 2), the (type, ordinal) attributes establish the top-level type. The category, subtype and description attributes (originated by us) provide further context.

Thus, for example, a data subject’s undergraduate degree is classified under (type:personal, category: education, subtype: qualification). The schema has a required attribute to denote which attributes must be present for a schema to apply. For example, if an IoT camera provides a timestamp, bitmap and light reading, only the bitmap attribute is required for the data to be treated as personal.

The schema is extended for secondary (i.e., inferred) types, to specify the conditions that must be satisfied to make an inference possible (Table 3). We do not claim that our schema is capable of classifying all possible personal data types, but, in the absence of a universal standard, it is sufficient for our purposes.

We currently define two types of conditions: (i) attributes—the additional set of personal data items that, when combined, could lead to a new inference and (ii) granularity—the threshold sampling frequency required to make an inference. When multiple attribute and/or granularity conditions are combined, all must hold for an inference to be satisfied. Finally, our status attribute distinguishes between personal data where (i) an inference has been made, and (ii) the data is available to make inference possible. For example, browsing data and gender may be enough to infer whether an individual is pregnant (i.e., these two items combined make pregnancy inferable) but if a node makes an actual determination on pregnancy, then the resulting data is inferred.

## 5. Implementation and Overview of IDE

We now provide details on how the work we have described has been implemented in the construction of our web-based integrated development environment (IDE). Underlying our work is an assumption that the privacy constraints built into our applications will be respected at runtime, i.e., that there is a PET that can enforce any contractual arrangements entered into between a data subject and an application. To this end, the applications that are constructed in our IDE will run on the Databox platform [42]. The Databox platform utilises an app store, from which privacy preserving apps can be downloaded to local dedicated hardware that runs within a home and connects to IoT devices and cloud services. For the purposes of the work reported here, the critical features of the platform are:**1.** **Support for local data processing**. Databox promotes (but does not mandate) local processing over data transfer and external processing. This makes it possible to construct applications that either do not transfer data to an external service, or perform data minimisation, ensuring that the smallest amount of data is transferred for a specific purpose.**2.** **Access to multiple silos of IoT data**. Databox has a growing number of drivers that are able to collect data from IoT devices and online services and make them available to apps.**3.** **Enforceable contracts between data subjects and applications**. Apps built in the IDE may be published to the Databox app store, see [42]. When publishing an app, a developer is required to provide information to inform data subjects about the data that will be accessed by the app, and the purpose, benefits, and perceived risks that attach to the use of the app.

The IDE runs out of a web-browser, with both the frontend and backend written in Javascript. We utilised Node-RED [43] as a starting point for the IDE, as this is a popular community IoT development tool. We undertook a full rewrite of the Node-RED frontend and modified the backend to add in our new features. The following are the most significant differences between the Databox IDE and Node-RED:Datastore nodes and output nodes communicate with the Databox API to collect data.Applications are bundled with a manifest that specifies the resources that it will require access to.Nodes in the IDE provide a function that takes the node inputs as an argument and outputs a schema describing the output data; in Node-RED, nodes are oblivious to the details of the upstream nodes.

In relation to the third feature, we maintain two separate schemas for each node. The personal schema contains the personal characteristics of each attribute of the data. It includes the rules to determine the conditions under which new personal data might emerge. For example, a part of our Bluetooth sensor datastore’s schema has an entry as follows:


  {
        type: “personal”,
        subtype: “relationships”,
        ordinal: “secondary”,
        status: “inferable”,
        description: “bluetooth scan information can be used to infer social relationships”,
        required: [“payload.address”],
        conditions: [{
             type: “granularity”,
             granularity: {threshold: 300, unit: “seconds between scans”}
        }],
        evidence: [
             “https://dl.acm.org/citation.cfm?id=2494176”,
             “https://doi.org/10.1109/MPRV.2005.37”
        ]
  }

This states that relationship data can be inferred if the Bluetooth sensor scans obtain the mac address (payload.address) with a frequency greater than or equal to once every 300s. Note that the schema can also provide further details on how the inference can be made; in this case, it will be presented to developers when they construct applications that make use of the Bluetooth sensor. We also maintain a JSON-schema for each node; this provides *type* information, so that each node can inform downstream nodes of the structure of its output data. For example, the following is a snippet of the JSON schema for the Bluetooth sensor:


{ 
      name: { 
            type: ‘string’, 
            description: ‘the name of the node, defaults to \’sensingkit\’’ 
      }, 
      id: { 
            type: ‘string’, 
            description: ‘<i>[id]</i>‘ 
      }, 
      type: { 
            type: ‘string’, 
            description: ‘<i>sensingkit</i>‘ 
      }, 
      subtype: { 
            type: ‘string’, 
            description: ‘<i>${subtype}</i>‘ 
      }, 
      payload: { 
            type: ‘object’, 
            description: ‘the message payload’, 
            properties: { 
                  ts: { 
                         type: ‘number’, 
                         description: ‘a unix timestamp’ 
                  }, 
                  name: { 
                         type: ‘string’, 
                         description: ‘user assigned name of the device’ 
                  }, 
                  address: { 
                         type: ‘string’, 
                         description: ‘mac address in the form aa:bb:cc:dd:ee:ff’ 
                  }, 
                  rssi: { 
                         type: ‘number’, 
                         description: ‘received signal strength indicator’ 
                  }, 
            } 
      } 
} 


The ‘description’ field in the schema is used to display type information to the developer. Schemas are only used in the frontend to support application development. Because the schema output from a node is influenced by how the node is configured, every time a node’s configuration changes, all downstream nodes will re-calculate their schemas.

The schemas are used by the frontend to mark the flow of personal data through an application; each edge is marked with a summary of the personal data type (e.g., primary, secondary, sensitive). If and when an ‘export’ node—the node that exports data off the Databox—is connected to, the IDE will gather information to help the developer determine whether a DPIA is likely to be required. We provide further details on this in the next section. With each change in configuration, the recommendation will update to reflect the changes, thus supporting in vivo reasoning, where any changes to node configurations or connections will cause any changes to privacy implications to be immediately recalculated. This is in contrast to typical data analysis, which is commonly performed less frequently (often when applications are in the latter stages of production).

On the backend, all applications are saved as JSON node-RED compatible flow files, alongside an application manifest that describes the resources (sensors/storage) that the application will access as well as the purpose and benefits of the app, which the developer is prompted for at publication time. The application is bundled into a Docker container [44] and uploaded to a registry (referenced in the manifest file).

By associating sources of data with a schema, our work distinguishes itself from taint tracking (which typically assesses data sensitivity based upon its source, e.g., microphone, GPS, camera), by providing additional context for a developer to reason more fully about the nature of the personal data flowing through an app.

### 5.1. Using the IDE

To illustrate a basic example in the IDE, consider Table 4 which outlines the relevant parts of the accelerometer schema for the flows in Figure 3 and Figure 4.

In Figure 3, *p2* is output from the accelerometer to show that personal data (i.e., a data subject’s gender) is *inferable* from the *x,y,z* components of its data (it is semi-transparent to denote it is *inferable* rather than *inferred*).

Similarly, with the profile node, *i1* is output to show *fullname.* When these are merged in the combine processor, the output schema will contain the accelerometer’s *p2*, and the profile’s *i1*.

In Figure 4, the combine node is configured to only combine the *x* and *y* components of the accelerometer data with the profile data. Since *x,y* and *z* are all marked as required (Table 4) for a gender inference to be possible, the combine node’s output schema will only contain *i1* (and not *p2*). The IDE automatically recalculates and re-represents the flow of personal data whenever a node or edge is removed, added or reconfigured. As flows get more complex, this becomes invaluable. It helps the developer to quickly determine how changes in configuration affect the flow of personal data. The IDE also flags points in an app that may require further attention from the developer, e.g., when personal data is being exported off the box (i.e., connected to the *export* node).

### 5.2. Providing DPIA Recommendations

In this section, we demonstrate how the personal data tracking work that we have described above may be used to support due diligence. Consider a health insurance app shown in Figure 5. The app creates a health profile, using a health profiler node, from smartwatch ECG data and a user’s postcode. It also takes grocery shopping data which it passes into a function which will generate a score. The output from the health profiler and the function are combined and exported to generate a quote.

At the point where data flows to an output node, the IDE flags an exclamation mark to warn that personal data is being utilised (in this case, exported). When selected, a DPIA screening form (Figure 6) is presented, generated in part from information on the app. On the right-hand side of the form is a column where yes/no answers can be provided for each question. The questions are drawn from the nine criteria provided by A29WP (see Section 3). Where possible, the IDE extracts app-specific information to provide relevant context for each of the questions. With regard to the use of sensitive or personal data, for example, the IDE summarises which personal data is being processed; in this case, identity data, personal (shopping data) and sensitive (heart trace) data. With regard to the use of evaluation/scoring, the IDE presents those parts of the application where it does occur (i.e., where data flows into the health profiler node) or where it might occur (where shopping data flows into the function node). Because the application makes use of the health profiler node, the evaluation/scoring criteria is set to “yes”; the developer cannot set it to “no” without first deleting the profiler node, given that the profiler node unequivocally entails evaluation and scoring. Five of the criteria are only relevant if data is exported from a user’s PET (the Databox in our case) where it may be subject to further processing. In this example, because personal data is exported, the developer is asked an additional set of general questions about the nature of any follow-on processing. Note also, with regard to the use of systematic monitoring, that the IDE examines the sensors used to determine whether monitoring is feasible. In this case, the use of smartwatch ECG data could theoretically be used to monitor a user’s behaviour. This is therefore flagged for consideration by the developer. Regardless of whether or not a DPIA is subsequently carried out, the answers to these questions are published alongside the app to demonstrate that due diligence has been exercised.

## 6. Enabling Due Diligence

GDPR creates a practical imperative for developers to take reasonable steps ensuring that their applications are compliant with regulation and that they therefore implement Data Protection by Design and Default (DPbD) in their construction. The Data Protection Impact Assessment (DPIA) provides a key heuristic for enabling due diligence and bringing about DPbD. Adopting a privacy engineering perspective, we have sought to move beyond developing external guidance to build due diligence into the development environment itself in a bid to (a) help developers reason about personal data during the in vivo construction of IoT applications; (b) advise developers as to whether or not the design choices they are making occasion the need for a DPIA; and (c) make specific privacy-related information about an application available to others implicated in application development and use (including data processors, data controllers, data protection officers, users and supervisory authorities).

We do not suggest that the IDE provides a silver bullet to the manifold challenges of compliance (e.g., providing compliance information to users and enforcing their rights), or that due diligence with respect to DPbD can be wholly automated. Of the nine criteria implicated in determining the need for a DPIA, five require that developers draw upon information external to app construction. Thus, criteria number three (where processing denies a right, service or contract), six (matching or combining datasets), seven (innovative new solutions), eight (data processed on a large scale) and nine (data concerning vulnerable subjects) turn on a developer’s knowledge of an app, its novelty, what it will be used for, its intended users and what might subsequently be done with their data. The IDE does not negate human skill and judgement, and in this respect there will still be need for education and learning (e.g., as to what constitutes ‘vulnerable’, or the ‘rights and freedoms’ that may be impacted through data processing). However, the IDE does sensitise developers to key DPbD concerns and makes us aware of the issues we need to be able to address if we are to satisfy ourselves and those to whom we are professionally accountable that we have acted with due diligence.

The IDE demonstrates that it is possible to build due diligence into application development not only by sensitising developers to core DPbD concerns but also by surfacing salient information and advice within the in vivo process of app construction. While the IDE cannot directly address criteria six (matching or combining datasets) or eight (data processed on a large scale), it can flag to the developer that personal data is exported elsewhere (i.e., for processing outside an app) and when a function node (i.e., a node that permits arbitrary code) operates on personal data. Furthermore, by introducing different classes of processing node, and specifically profiling nodes, the IDE can flag concerns that directly relate to DPIA criteria number one (evaluation and scoring) and two (automated processing with legal or similar effect). Criteria number four (systematic monitoring) is made visible by reference to the type and rate of flow of personal data, both of which the IDE can determine from an app’s construction (i.e., when personal data flows to an output at a threshold rate). Take, for example, a camera that continuously streams data to a screen or to the cloud. This will clearly create conditions where systematic monitoring is feasible; a camera configured to only take a picture every hour, however, may not. Again, the IDE has access to the personal data types and configurations of input nodes (i.e., sensors) and is able to utilise these to help make this determination. The characteristics of data flowing through an app also enable the IDE to assess whether criteria number five (sensitive data or data of a highly personal nature) is a relevant concern present this to the developer.

Our IDE has been inspired by the program analysis community, particularly the use of taint tracking. However, unlike taint tracking, our goal is not to discover program vulnerabilities but to make the flow of personal data amenable to analysis by developers so that they can make demonstrably diligent design decisions with respect to the DPbD requirement. Just as taint tracking has its limitations, so does the IDE: our schema will not cover all possible personal data types, we cannot encode an exhaustive set of rules for all possible inferences across all personal data, and we cannot definitively state which DPIA criteria are met; we can only prompt the developer to assess potential conflicts. However, privacy assessments are unlikely ever to become fully automated, given the highly interpretable character of the law and the limits of mapping between technical and legal domains. Nonetheless, encouraging early reasoning that is directly relevant to the *application in hand* is in the spirit of regulation and manifestoes such as Privacy by Design. The IDE demonstrates that we can move DPbD beyond costly expert advice, disconnected documentation, and after the fact considerations. Instead, we might embed due diligence in the application development environment and in the actual in vivo process of app construction to assist developers in understanding the privacy risks that accompany their particular design choices and to help them arrive at an informed judgement as to whether or not a DPIA is needed to comply with the requirements of GDPR.

Information on where to access the Databox and IDE can be found in ‘Appendix A’ section at the end of this paper.

## 7. Future Work

In this paper, we have argued that embedding tools within an IDE to aid a developer’s reasoning about personal data is an essential part of designing privacy into applications. Our work sits within a wider eco-system and assumes the existence of a home-based Privacy Enhancing Technology that is capable of (i) accessing a wide variety of personal data sources and (ii) enforcing an app’s privacy obligations. Both of these must be met in order to encourage developer communities to build apps in earnest. Once the technology has matured (the Databox is one such solution that supports a limited, but growing variety of sources of personal data), we will be in a position to undertake a meaningful user evaluation in order to understand more fully how our tools are used by developers (where genuine privacy tradeoffs can be made during app design), and the extent to which they help improve upon privacy related design decisions.

## 8. Conclusions

IoT application developers, like developers everywhere, find themselves confronted by the General Data Protection Regulation (GDPR) and the associated Data Protection by Design and Default (DPbD) requirement. It is imperative that developers demonstrate due diligence and take reasonable steps to meet the DPbD requirement when building IoT applications. However, developers largely lack the competence to understand legal requirements or to translate them into technical requirements, and legal experts are often similarly challenged in reverse. We have presented an integrated development environment (IDE) to demonstrate the possibility of building due diligence into IoT application development. The IDE is designed to help developers understand personal data and reason about its use, to identify potential DPbD conflicts, and to make specific design choices available to others to whom the developer is professionally accountable. It flags the privacy implications of the design choices a developer makes as they make them and provides them with the tools to reflect upon and alter a design decision when it is most easily accomplished. The IDE is not a silver bullet, but it does show that is possible to engineer privacy into the IoT. Though we have concentrated on an application environment for home IoT, the principles of our work, in particular the in vivo reasoning about privacy implications at development-time, may be applied to other development environments whose products process personal data (for example, the Android/iOS mobile development environments).

## Figures and Tables

**Figure 1 sensors-19-04380-f001:**
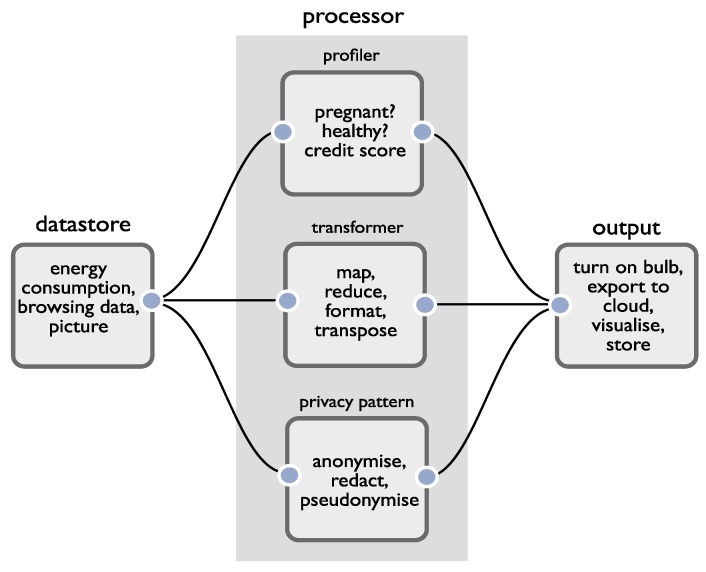
Integrated Development Environment node types.

**Figure 2 sensors-19-04380-f002:**
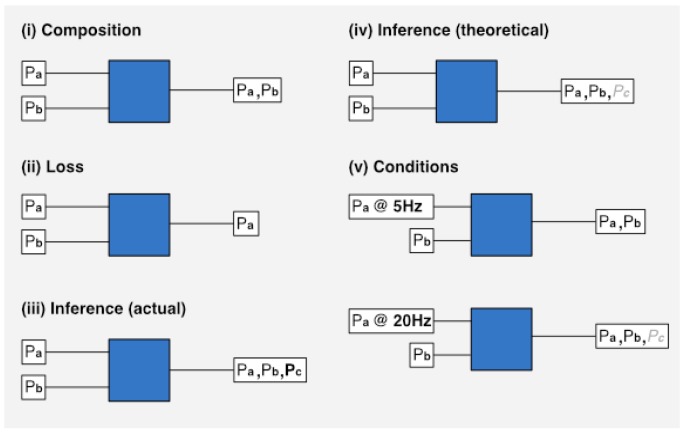
Personal data flows.

**Figure 3 sensors-19-04380-f003:**
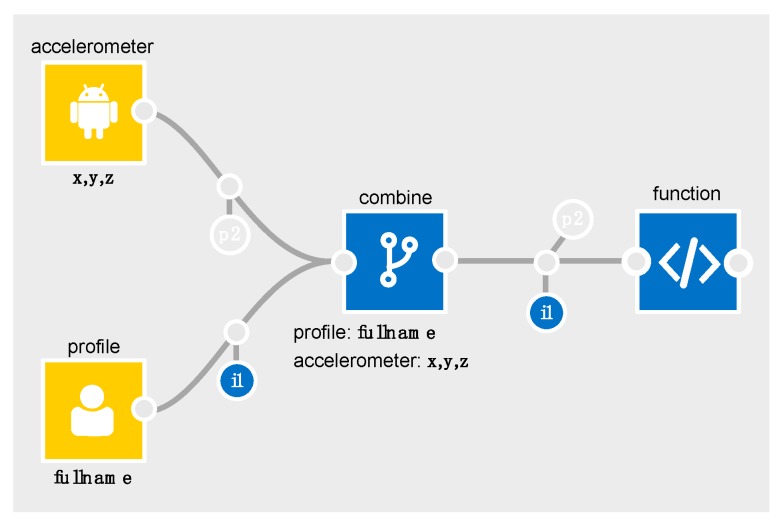
Applying the schema (a).

**Figure 4 sensors-19-04380-f004:**
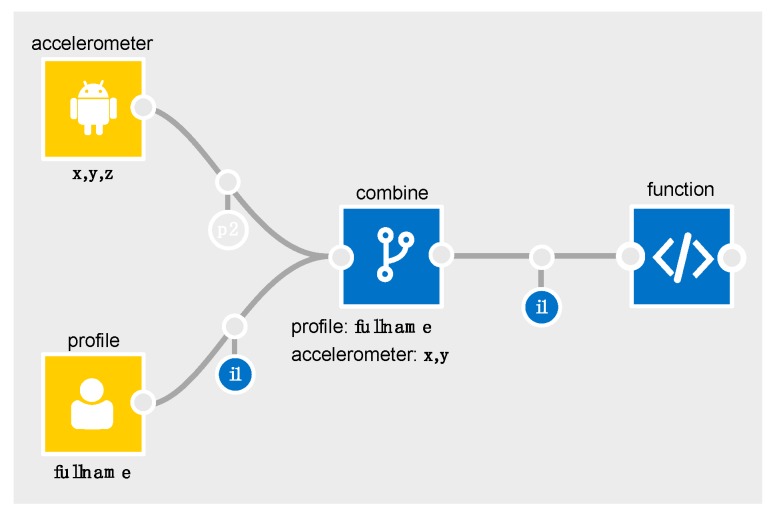
Applying the schema (b).

**Figure 5 sensors-19-04380-f005:**
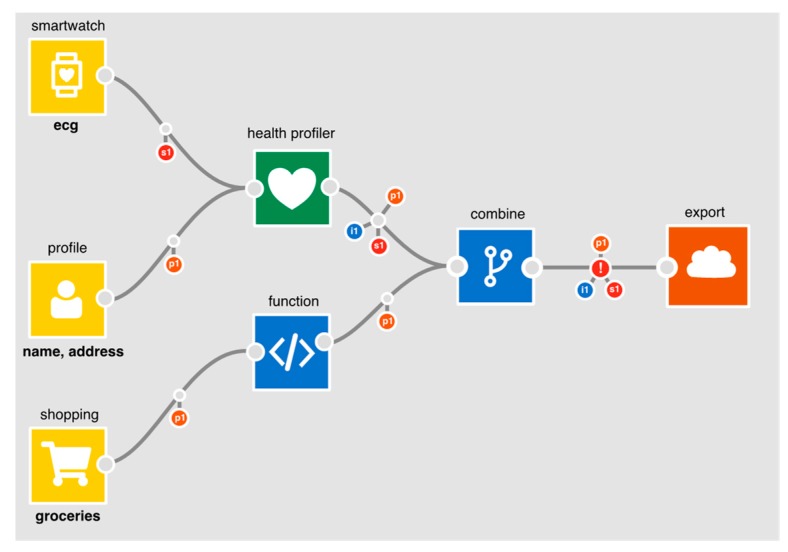
Health Insurance quote app.

**Figure 6 sensors-19-04380-f006:**
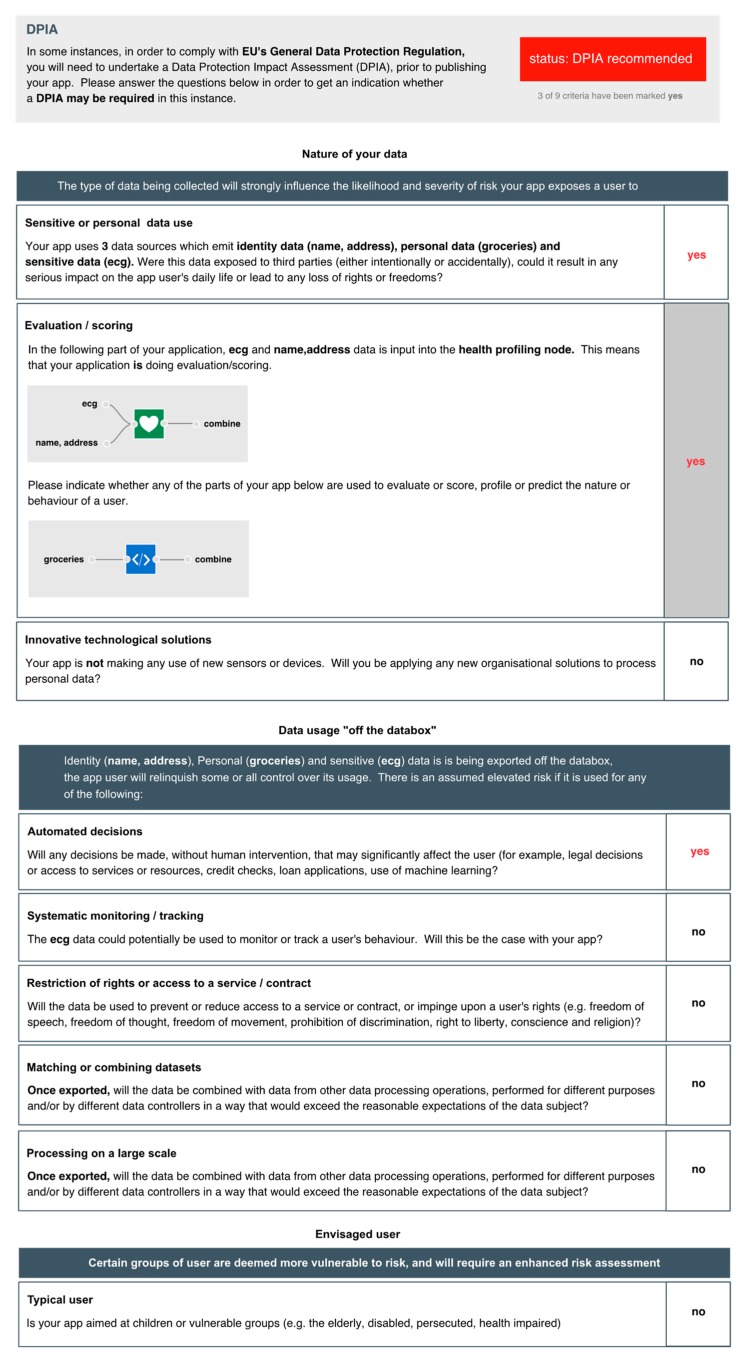
IDE DPIA recommendations.

**Table 1 sensors-19-04380-t001:** Personal data types.

Label	Type	Ordinal	Description
i1	identifier	primary	data that directly identifies a data subject
i2	identifier	secondary	data that indirectly identifies a data subject
p1	personal	primary	data that is evidently personal
p2	personal	secondary	derived personal data
s1	sensitive	primary	GDPR special categories of data
s2	sensitive	secondary	derived sensitive data

**Table 2 sensors-19-04380-t002:** Personal data attributes.

Attribute	Description
type	identifier|sensitive|personal
ordinal	primary|secondary
category	physical|education|professional|state|contact|consumption|…
subtype	e.g., physical includes hair colour, eye colour, tatoos etc. education includes primary school, secondary school, university etc.
description	details of this particular item of personal data (and method of inference if secondary)
required	list of attributes of this data that must be present in order for it to constitute as personal

**Table 3 sensors-19-04380-t003:** Secondary data attributes.

Attribute	Description
confidence	an accuracy score for this particular inference, ranging from 0 to 1
conditions	list of *granularity|attributes.*
evidence	where possible, a set of links to any evidence that details a particular inference method
status	inferred|inferable

**Table 4 sensors-19-04380-t004:** Accelerometer personal data schema.

Attribute	Description
type	Personal
subtype	Gender
ordinal	Secondary
required	[x,y,z]
conditions	Type: granularity, threshold:15, unit: Hz

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
