# Peer review of "Privacy Engineering for Domestic IoT: Enabling Due Diligence"

_sensors, 2019, doi:10.3390/s19204380_

Round 1

Reviewer 1 Report

The paper is very well written and authors clearly know their subject. The topic how to improve privacy by design for IoT is of utmost importance. The major issue with the paper that the claims of the authors:

improved privacy awareness of the designer improved privacy of the resulting implementation

are not evaluated.

I admit that in principle their claims are plausible as the tool presented forces the designer to think about privacy issues. But the paper lacks a "user study" that provides feedback form designer who used the tool for realizing a certain application saying "now I know what I need to consider" as well as a study that analyses applications realize with and without use of the tool concerning the level of privacy protection. 

In addition the authors highlight as a drawback of existing tools that they require a data flow analysis for privacy support. But if I got it right their tool requires the designers to do similar steps if not the same. This definitely needs clarification.

Reviewer 2 Report

Nowadays, not many software developers realize what GDPR compliance entails and how it relates to software development. Even for the ones who understood the implications of GDPR compliance, another problem arises: how to embed the GDPR requirements in the software development to produce compliant applications.
This paper presents an IDE designed to help developers identifying potential GDPR conflicts. As the authors claim, this IDE is not the silver bullet but will help for sure the developers without the legal knowledge building applications compliant with GDPR specifications.
The machine learning is, in my opinion, one of the most severe threats to privacy because the algorithms are getting better results with fewer data. For this reason, the produced recommendation about DPIA can be wrong. In my opinion, the authors should also assess the impact of machine learning algorithms in the proposed solution.
The IDE should be available to the community to be tested and validated.
Can this IDE be integrated into the most used Android and IOS development environments? Build the function prototypes should be enough.

Round 2

Reviewer 1 Report

The authors addressed my comments in a satisfactory way. So, in my view I recommend to publish the paper in its current version.